# Full-Time or Working Caregiver? A Health Economics Perspective on the Supply of Care for Type 1 Diabetes Patients

**DOI:** 10.3390/ijerph19031629

**Published:** 2022-01-31

**Authors:** Sayaka Sakoda

**Affiliations:** Japan Society for the Promotion of Science, Kyoto University, Kyoto 606-8501, Japan; sakoda.sayaka.22c@st.kyoto-u.ac.jp

**Keywords:** type 1 diabetes mellitus, socio-economic status, government aid, caregivers

## Abstract

Type 1 diabetes mellitus (T1DM) is a chronic disease requiring lifelong insulin treatment. T1DM patients require care given not only by themselves but also by their family members, particularly in childhood-onset cases. This study aims to identify the relationship between health expenditure, HbA1c and other health outcomes and the socio-economic status of patients and their families, with a focus on family employment status, i.e., whether the caregiver is employed or is a homemaker. To clarify the relationship between the level of health, such as expenditure on health care and HbA1c, and the socioeconomic status of patients and their families, we focus on whether they are “potential full-time caregivers”. Using this analysis, we estimated the hypothetical health care expenditure and HbA1c and showed that male patients have higher expenditure and lower HbA1c when their caregiver is a potential full-time caregiver, whereas younger female patients have higher health care expenditure and lower HbA1c when their caregiver is employed. This finding is not meant to serve as criticism of health care policy in this area; rather, the aim is to contribute to economic policy in Japan for T1DM patients 20 years and older.

## 1. Introduction

Type 1 diabetes mellitus (T1DM) is a chronic disease characterized by an absolute insulin deficiency resulting from the progressive immune-mediated destruction of pancreatic islet β cells. T1DM depends on “luck”, which occurs through an autoimmune mechanism in young people until puberty. Management of T1DM requires many lifelong daily tasks, e.g., glucose monitoring, adherence to insulin regimen, and meal planning that the child and/or family must perform to maintain a healthy metabolism and glycaemic control [1]. The main treatment methods are multiple daily insulin injections (MDI), continuous subcutaneous insulin infusion (CSII), and sensor-augmented pump (SAP). Except with CSII and SAP, automatic glucose measurement and self-monitoring of blood glucose must take place before insulin injections. Particularly when the patient is a child, parents will function as care coordinators, alongside health professionals and system supporters in addition to their regular parenting duties.

Many studies report a strong socio-economic gradient in health, regardless of the population studied and the method of measurement [2,3,4,5,6,7,8,9,10]. Regarding T1DM, besides factors relating to poor adherence to diabetes treatment, socio-economic status (SES) and other family demographic factors such as maternal education and knowledge, rural residence, and distance from the clinic influence HbA1c (the standard measure used for assessing long-term glycaemic control) [11,12,13,14,15,16,17,18,19,20,21] are of the opposite argument, in that the causality runs from health to SES, as in recent empirical studies by [22,23]. The diagnosis of chronic disease in a child affects not only the patient but also the child’s family [24,25]. This may lead to a deterioration in the family’s employment situation and a reduction in income, for example, if the patient is unable to work or attend school or if the patient’s parents must leave their jobs to look after the patient. Previous research has clarified that T1DM in childhood has a negative effect on labour market outcomes later in life, which means the causality instead runs from health to SES [26]. 

In Japan, the “Child Chronic Specified Disease Treatment Project” based on the Child Welfare Law covers medical expenses for T1DM until the child turns 20 years old. Afterwards, patients bear the costs of all daily treatment. Any change in the treatment plan may also have a negative impact on the patient’s health and socio-economic status. It is difficult for paediatric patients with T1DM to inject insulin independently, and they require parental or guardian care. The literature has found that parental and spousal recommendations are important in the demand for better medical equipment, not only for paediatric patients but also for adult patients [27]. Previous studies have identified the causal impact of SES on health levels and health levels on SES. However, they have not investigated the impact of the SES of caregivers. In studies of inequality, it is common to consider SES background at 15 years of age, and this paper employs this approach to studying T1DM. To address this, this study focuses on the impact of SES on health status and vice versa, with particular attention to differences in the SES of caregivers of patients with T1DM in Japan. We also estimate the hypothetical value of the impact of either health care funding or caregiver presence on blood glucose levels as measured by HbA1c. We use these results to suggest directions for health care funding for patients with chronic disease in Japan.

Focusing on Japan, the number of children requiring daily medical care (usually called “medical care children” in Japan) is increasing each year, with 18,272 cases in 2016 [28], especially for parents, who are the primary caregivers of children in medical care, and the load on mothers is particularly high. Despite this, in Japan, most studies have focused on the employment of patients themselves, and there are fewer studies concerning the employment of the caregivers of paediatric patients than in other countries. There is a lack of basic national data on the employment status of families with children in medical care. There are no concrete measures to support parents’ employment. 

From this point of view, one representative study is [29], in which a survey was conducted regarding 639 mothers of adults with intellectual disabilities. It reported that the employment rate of women in their 50s was 39.8%, compared with 61.5% for married women in general. Another study [30] also revealed that the employment rate of women with “medical care children” was only 42.7%. Among working mothers, 53.1% were engaged in a professional or technical job [30]. This reflects the higher costs to care for medical care children directly or indirectly, with an alternative strategy for mothers of paediatric patients allocating money earned through working in a high-income occupation. 

Figure 1 illustrates the flowchart of causality running from SES to health level and health level to SES. Regarding Figure 1, does the inactivity of the patient and their family’s work lead to a lack of access to high-quality medical treatment? or does it lead to maintain health condition by concentrating on care? To solve this puzzle, we need to focus on the labour status of the patient and their family. 

In this paper, we attempt to evaluate the following hypothesis regarding mothers or spouses (both wife and husband) becoming full-time homemakers or caretakers, i.e., “potential full-time caregiver”. To respond to this question, we obtain counterfactual out-of-pocket medical expenditure and HbA1c, especially shedding light on labour status of the patient’s caregiver after using regression analysis to clarify the socio-economic factors of determining out-of-pocket medical expenditure and HbA1c.

The rest of the paper is organized as follows. Section 2 explains the materials and methods. Section 3 provides the regression results for out-of-pocket medical expenses and HbA1c and counterfactual outcomes for the caregiver. Using these results, we then estimate the counterfactual HbA1c, given health care funding and caregiver availability. Section 4 discusses health care subsidies and economic policies for T1DM in Japan, and Section 5 is the conclusion.

## 2. Materials and Methods

### 2.1. Materials

In this section, we explain about the materials, e.g., the design, setting and population under the study. 

There are also a few surveys [31,32,33] conducted on the socio-economic aspects of T1DM patients. However, it is not possible to study the socio-economic background of the patients and their families and the effect of the treatment because of the following reasons.

Ref. [34] provides descriptive statistics on the status of patients who received the “Child Chronic Specified Disease Treatment Project” as of 2010; however, it does not include patients developing the disease over 20 years old. Ref. [35] also provides the study based on the survey for not only under-20-years-old patients but also for over-20-years old patients; however, it does not contain the feature of disease, e.g., onset age, seriousness, and socio-economic background of patients and their parents, since the survey is conducted through the patient’s physician. 

We administered an original questionnaire survey to T1DM patients and their families in Japan over the Internet during the period from 4 November to 2 December 2016, by collaborating with the largest association of T1DM patients in Japan, the Japan IDDM Network, and a network management company, NTT Com Online Marketing Solutions Corporation (hereinafter “NTT Com”) to ensure sufficient sample size. We used information on the monitors registered with NTT Com. In total, 947 respondents participated in the survey, but only 410 responses were valid, as we restricted the sample to those using medical devices such as MDI, CSII, and SAP. The information gathered comprises demographic (age, sex, and place of residence), socio-economic (marital status, employment status, and income level), and health and health economic (age at onset of T1DM, diabetic complications, treatment choice, out-of-pocket medical expenses, frequency of hospital/clinic visits, and government financial aid) variables. While suitable epidemiological surveys exist as is mentioned above, this is the first detailed socio-economic survey of T1DM patients in Japan, making the survey findings in themselves a significant contribution to the literature.

As mentioned, in Japan, the cost of medical care up to the age of 20 declines because of the “Grant-in-Aid Program for Chronic Diseases in Childhood”. This system reduces out-of-pocket expenses for children with target diseases, including T1DM. Thus, patients younger than 20 are eligible for aid through which the co-payment rate reduces to 20%, subject to an upper bound on the monthly out-of-pocket payments. This varies according to parental household income to a maximum of JPY 10,000 (USD 91, assuming an exchange rate of USD1 = JPY 110). In summary, an eligible patient pays the lower amount of the 20% co-payment and the out-of-pocket cost limit. This financial aid for the patient ends at the age of 20. As no similar government support exists for adult T1DM patients, they must pay the usual 30% co-payment. Table 1 summarizes the typical out-of-pocket expenses for a T1DM patient (based on age). The “Special Child Rearing Allowance” provides governmental income support for parents who look after their disabled children at home. A T1DM patient younger than 20 is eligible for second-degree disabilities. As of 2018, parents receive a monthly allowance of JPY 34,430 (USD 313). Given the costs involved and the low level of government assistance, a diagnosis of T1DM is then increasingly responsible for catastrophic expenditures that push patients and their families into a medical poverty trap.

The out-of-pocket medical expenses include the medical costs for treating T1DM and its complications. The independent variables include household income, onset timing, the frequency of outpatient service use, the number of complications, a time discount rate, a female dummy, a care dummy by age range, and government aid. Table 1 provides the details and descriptive statistics for the independent variables. 

We briefly describe the epidemiological findings arising from the descriptive statistics, which is shown in Table 1. Table 1 contains 2 brackets; one is for continuous variables and the other is categorical variables. 

The upper bracket of Table 1 contains the descriptive statistics for continuous variables. The explanation of each continuous variable is below. First, out-of-pocket medical expenses are the amount of out-of-pocket expenses for IDDM treatment and complications during 2015. The amount was paid at the medical institution, even if patients received a refund due to “High-cost Medical Expense Benefit”. This “High-cost Medical Expense Benefit” can be claimed if the cost-bearing limit of medical expenses exceeds the specified amount for the month. The total amount (30% of medical costs) is paid first, and a refund is later received. In the questionnaire, we asked, “Please provide the amount of your out-of-pocket expenses for IDDM treatment and complications over the last year. (*Please enter the amount you paid at the medical institution, even if you received a refund due to high-cost medical care expenses, etc.)”. Second, household income is the amount of household income in 2015. Lastly, HbA1c is the latest HbA1c value (NGSP value), and we asked, “Please provide us with the most recent HbA1c value (NGSP value) of patient”.

The lower bracket of Table 1, however, contains the descriptive statistics for categorical variables. 

In the questionnaire, we asked whether respondents have the following complications: diabetic retinopathy, diabetic nephropathy, neuropathy, gangrene, coronary artery disease, cerebrovascular disease, other. As for complications, most patients do not experience any complications.

Time discount was obtained from the question “*How much would you be happy to receive as a minimum after 13 months instead of JPY 10,000 (**USD 91, assuming an exchange rate of USD 1 = JPY 110**) after one month?*” shows the discount placed on returns receivable or costs payable in the future. The use of a high discount rate implies that people place less weight on the future and that they are impatient. Most of the time discount rate is under 10%, which means that they place high weight on the future. 

As for the varying timings of onset, around 48.7% of patients experience the onset of T1DM after age 30, which is consistent with [33]. Those who had an onset over the age of 20 were unable to be captured by previous surveys in Japan, e.g., [34,35], which aimed to track juvenile onset and “Grant-in-Aid Program for Chronic Diseases in Childhood” users. 

The most frequent use of outpatient services is once per month. 

Next, we shift to the explanation of the variable of caregiver of patients. As for juvenile patients, it has been pointed out that 97.5% of children with disabilities at home are primarily cared for by their mothers, and that children with medical care place a heavy load on mothers [36,37]. It is clarified that patients with a spouse demand newly developed medical equipment in [27]; thus, we focus on patients’ mothers and spouses as caregivers, which is categorized as “care dummy” in this article.

We specify the care dummy by age range. If the patient’s mother or spouse, for both wife and husband, is a full-time homemaker or not working, the caregiver dummy takes a value of one, otherwise zero. Hereinafter, we define this caregiver as a “potential full-time caregiver”.

In the field of labour economics, there is a socio-economic bias when engagement of a patient’s mother or wife in paid work is related to their own and husband’s economic level, a tendency recognized as “Douglas–Arisawa’s second law”. The rule of thumb is that the higher the head of household’s income, the less likely the spouse has a job. However, theoretically, we can dismiss such a bias because of the random onset of T1DM. As can be seen from the descriptive statistics, the labour status of the mother or spouse of a patient under 20 years old indicates low labour force participation. For single patients under 20, 63.46% of mothers work. This result is consistent with previous studies, such as in [30]. 

Lastly, we also focus on those who receive governmental aid. This “government aid dummy” takes a value of one if the patient under 20 years old receives at least one of the “Grant-in-Aid Program for Chronic Diseases in Childhood” or “Special Child Rearing Allowance”; otherwise, it is zero. For example, for adult recipients who have received some government aid in the past, the dummy variable has a value of zero. 

### 2.2. Methods

Generally, experiments are difficult in the field of social sciences. However, in recent years, methods have been developed to predict social phenomena by assuming hypothetical values. The most popular methodology is a randomized controlled trial, known as RCT, which randomly divides a target group (e.g., patients with a particular disease) into several groups (e.g., treatment groups and control groups) and measures and clarifies the impact/effect of the experimental intervention. 

In this paper, to see how the caregiver’s employment status determines out-of-pocket medical expenditure and HbA1c, we have to estimate how hypothetical marginal effects of becoming a “potential full-time caregiver” increase out-of-pocket medical expenditure or worsen HbA1c. 

Upon the discussion above, caregivers should be randomized into the two groups by their employment status (e.g., “potential full-time caregiver” vs. no “potential full-time caregiver”) to treat confounding effects; however, it is unavoidable given the study design since interventions for working and labour status hardly happen. To curb this problem as much as possible, this paper employs the margin effect analysis which assumes hypothetical values.

Based on Equation (1), we run the regression for out-of-pocket medical expenditure and HbA1c. 

$y_{i}$ is out-of-pocket medical expenditure and HbA1c, and $x_{i}$ contains household income, onset timing, frequency of outpatient visits and number of complications, time discount rate, care dummy by age range and government aid dummy. The subscript *i* indicates an individual, and ε is the error term.
(1)$$y_{i}=\alpha +\beta_{i}x_{i}+\varepsilon$$

We provide the results for a divided sample (male and female). First, we estimate out-of-pocket medical expenses, and then HbA1c.

After conducting the regression, we estimate the counterfactual out-of-pocket medical expenses and HbA1c based on the caregiver’s labour status. 

To obtain counterfactual out-of-pocket medical expenses and HbA1c, we use marginal effect as is described in Equation (2).
(2)$$\frac{\partial y_{i}}{\partial x_{i}}=\beta_{i}$$

Especially, we try to obtain predicted value applying the effect of caregiver’s labour status when government aid exists or not and marginal effects of the independent variable $x_{i}\;$ constituting the interaction. Then, we estimate the four predicted values. 

To answer the research question, we estimate the marginal effect on:The out-of-pocket medical expenditure of care dummy by age;HbA1c of care dummy by age range;HbA1c of care dummy and government aid by age range.

The first values show which caregiver’s labour status affects the out-of-pocket medical expenditure. The second values display which caregiver’s labour status affects HbA1c, and the last values show which caregiver’s labour status affects HbA1c with/without governmental aid.

For the actual analysis, this paper used the margins command in Stata following [38].

As is noted above, even when using the margins command to calculate hypothetical values, there are confounding effects present. 

## 3. Results

This section explains the results of the regression for out-of-pocket medical expenses and HbA1c and then those for the marginal effects of the care dummy. After these estimations, we obtain the marginal effect on HbA1c of care dummy by age range and government aid to see how caregiver’s labour status affect HbA1c under the exist/absence of government aid. 

Table 2 and Table 3 provide the results relating to out-of-pocket medical expenses, with those for HbA1c in Table 4, Table 5 and Table 6. Table 7 shows the marginal effect on HbA1c of care dummy by age range and government aid. 

To start, we discuss the regression estimates and marginal effects for out-of-pocket medical expenses, with Table 2 providing these by subgroup. Model 1 shows the results for the whole sample, with Models 2 and 3 for the male and female patient subgroups, respectively. 

Table 2 shows the regression results of out-of-pocket medical expenses. 

Model 1 presents significant results focusing on the whole sample, household income, the frequency of outpatient use, especially once or few times per week/month, the time discount rate, the female dummy, 40–49 years old and working caregiver, and government aid. This means that higher-income households incur higher out-of-pocket medical expenses. By contrast, there are significant negative effects for the government aid dummy. A lower frequency of outpatient visits reduces out-of-pocket medical expenses, while a high time discount rate indicating that patients tend to be impatient is also positively related to out-of-pocket expenses. The caregiver dummy by age range, 40–49 years old, and working caregiver exhibit negative estimates. 

Models 2 and 3 show similar findings. However, what is interesting is the value for the care dummy by age range. Among results for male patients, a patient who is 10–19 years old and has a “potential full-time caregiver” has a significant positive value. By contrast, for female patients, some categories of the care dummy by age range exhibit significant negative estimates. Table 3 details the marginal effects of the care dummy by age range. 

Table 2 shows the predicted value on the out-of-pocket medical expenditure of care dummy by age. 

In terms of counterfactual medical expenses, that of “potential full-time caregiver” suggests higher expenditure for both 20–29-year-old male patients and 40–49-year-old female patients whose spouse or mother is at least a full-time homemaker. Outside these two categories, the counterfactual out-of-pocket medical expenditure when the caregiver works displays a higher value. In the analysis above, the caregiver variable exerts a significant effect for female patients, and attention is given to this in the following discussion.

Middle-aged patients with a “potential full-time caregiver” incur higher out-of-pocket medical expenditures. We suggest two reasons: direct and indirect. A “potential full-time caregiver” may have the capacity to care regarding a patient’s health condition and frequency of outpatient visits. Conversely, middle-aged patients are eager to maintain their health condition to support their family. Nonetheless, what matters is not out-of-pocket medical expenditure, but rather the level of the patient’s health. 

Next, we refer to the regression estimates and marginal effects where HbA1c serves as the dependent variable. Table 4 and Table 5 provide the results for estimated HbA1c for male and female patients using Models 4 and 5 and Models 6 and 7, respectively. 

As shown in Table 4, for male patients, one complication, the time discount rate, and the government aid dummy show significant estimates. To focus on the care variable, we add Model 5. In Model 5, the care dummy displays significant negative results, which indicates that a full-time homemaker/mother leads to an improved HbA1c level.

Table 5, which provides the estimation results for female patients, the onset timing especially for 10–19 years and 30–49 years, the frequency of outpatient visits, two or more complications, and 20–39-year-old patients with a “potential full-time caregiver”, exhibits significant results. Focusing on the onset timing, we can confirm that adolescent onset leads to rising HbA1c, especially for female patients.

The results in Table 6 provide two interesting points. One is that with paediatric patients, having a “potential full-time caregiver” leads to a lower HbA1c, which we can clarify by comparing Cases 1 and 2 and Cases 3 and 4. Another is that for all other ages except females aged 40–49 years old, “potential full-time caregiver” is related to a higher HbA1c level. 

In the youngest age group, homemakers have a better HbA1c, but in the adult group, HbA1c is worse. We suggest that this is because of governmental subsidies for paediatric health care. HbA1c may be lower in younger people with less financial burden and who can provide better care. This may be because the financial burden is lower in the younger age group because of subsidized health care, and the presence of homemakers allows them to take better care of their children, resulting in a lower HbA1c. Finally, in this section, we look at the effect of caregiver employment status and health care subsidies.

Table 7 shows the predicted value aiming at how caregiver’s labour status affects HbA1c by the case of whether the governmental aids exist or do not exist. 

The results in Table 7 provide three main findings. First, for patients under 40 years old and female, a “potential full-time caregiver” is superior to government aid in lowering the level of HbA1c. Second, except for this age group, having a working caregiver leads to a better HbA1c than having a “potential full-time caregiver”. Finally, for all ages, there are significant differences for male and female patients. For male patients, the presence of a caregiver rather than government aid is more important in maintaining HbA1c. For female patients, a working spouse or mother is important for maintaining the HbA1c level. However, the difference is smaller than for male patients. 

## 4. Discussion

The purpose of the analysis conducted in this study was to gain a better understanding of the correlations between the labour status of caregivers of diabetic patients and out-of-pocket medical expenditure, HbA1c and governmental aid. 

Studies on the issues of labour status associated with providing care to patients with diabetes and factors influencing out-of-pocket medical expenditure and HbA1c considering governmental aid have not been conducted in Japan before.

In the previous section, we conducted a regression analysis of health care expenditures and HbA1c by considering not only the patient but also the SES of the caregivers, and based on the results, estimated counterfactual health care expenditures and HbA1c given the employment status of caregivers. We also estimated the counterfactual out-of-pocket medical expenditure and HbA1c based on caregiver working status. 

We found that the impact of caregivers on counterfactual out-of-pocket medical expenditure was negative for younger age groups when the patient was female, i.e., out-of-pocket medical expenditure was lower. Based on the analysed counterfactual out-of-pocket medical expenditure, middle-aged patients with a full-time house-caregiver incur higher out-of-pocket medical expenditures. We explain this finding both directly and indirectly. Directly, a full-time house-caregiver may have more allowance to care about a patient’s health condition and the frequency of outpatients. Indirectly, middle-aged patients are eager to maintain their health condition to support the patient’s family. In terms of HbA1c, the full-time caregiver dummy is significant for all age groups for men and 20–39-year-old women. Based on these results, we estimated HbA1c and found that for women under 40 years old, having a “potential full-time caregiver” rather than government support was associated with lower HbA1c levels. However, this result is not always valid in the case of long-term data analysis, since it is referred that families that experience reduced financial resources because of lost income from either a cut in work hours or having to stop work were perceived to be a more significant financial burden, and often responded that they needed additional income to cover medical expenditures [39]. 

The expenditure amount may be relatively small compared to other household expenses; families with children who need special health care may be financially stretched because of many years of persistent costly care resulting in lower expenditures causing the perceptions of problems [40]. 

It is also referred that meeting the fiscal demands of health care expenditures with decreased financial resources may result in families selling assets, taking out loans or mortgages, using savings, or seeking additional employment [41]. Re-establishing financial well-being may be hindered by decreased employability and loss of career advancement and mobility caused by being out of the workplace for any extended period to care for the child [42]. Taken together with the regression analysis results, this effect is likely to be more vital for those aged between 20 and 40. In combination with the significant consequences for the regression analysis (Model 5) for all age groups of men, the presence of an unemployed person at home contributes to lowering HbA1c and maintaining health. Finally, we estimated a hypothetical HbA1c based on the caregiver’s employment status and the availability of health care subsidies. The results show that for men, the presence of a caregiver is more critical for maintaining HbA1c than government assistance, with a difference of about two. For women under 40 years of age, the presence of a working caregiver rather than government assistance can lead to lower HbA1c levels. In other words, the presence of a caregiver who can afford to always care for a patient is linked to the level of health for men and women 40 and older, whereas the presence of a caregiver working and supporting the patient financially is linked to the level of health for women under 40. The reduction in HbA1c levels brought about by the presence of a caregiver is also greater among younger age groups. As discussed, Japan’s only policy for people with T1DM is to subsidize medical costs up to the age of 20. It is shown that removing this subsidy has led to distorted demand for medical equipment [27]. Nonetheless, they have also shown that family recommendations strongly affect the demand for medical equipment. This means that the presence of caregivers has a significant impact on the level of health of people with T1DM. 

## 5. Conclusions

This study focused on the impact of SES on health status and vice versa, with particular attention to differences in the SES of caregivers of patients with T1DM in Japan. Using an original survey conducted in 2016 targeted at Japanese T1DM patients, we studied the following three points:We ran the regression on out-of-pocket medical expenditure and HbA1c based on the patient’s and caregiver’s socio-economic factors, especially the caregiver’s labour status.Based on the above results, we estimated counterfactual out-of-pocket medical expenditure and HbA1c.We also estimated a hypothetical value of the impact of either health care funding or caregiver presence on blood glucose levels.

When we looked at differences in caregiver employment status, we found that the effects varied by sex and patient age. As is noted in Section 2, to obtain the true effect of caregiver’s employment status, a randomized control trial should be applied; however, it is unavoidable given the study design, and we employed marginal effect analysis for this attempt, which means there are surely confounding effects present which should be taken into consideration. The overall finding is that counterfactual HbA1c is more influenced by caregiver employment status than by government funding. For women under 40 years old, HbA1c was lower when the caregiver was employed. For the rest of the population, men and women 40 and older having an unemployed caregiver resulted in a lower HbA1c. As discussed, this conclusion is not for the purpose of denying the benefit of government aid such as the “Grant-in-Aid Program for Chronic Diseases in Childhood” and “Special Child Rearing Allowance”. These governmental aids are available only to paediatric patients, and they have time restrictions. In future research, we will assess how continuous economic policy relating to T1DM patients would be appropriate. 

The conclusions of this paper do not negate the need for health care funding. This is because medical devices such as SAPs and CSIIs, which lower HbA1c, have either not yet penetrated the Japanese health care system or are not used because they interfere with daily life and require blood glucose control by oneself or family members. Therefore, if patients, their families, and doctors are informed about these medical devices, and if the quality of medical devices further improves, it is expected to be more effective than the presence of caregivers. Further, while there is a need for additional analysis, it seems that there is also a need for economic policies such as employment and wages for people with T1DM and their families, rather than time-limited health care subsidies.

## Figures and Tables

**Figure 1 ijerph-19-01629-f001:**
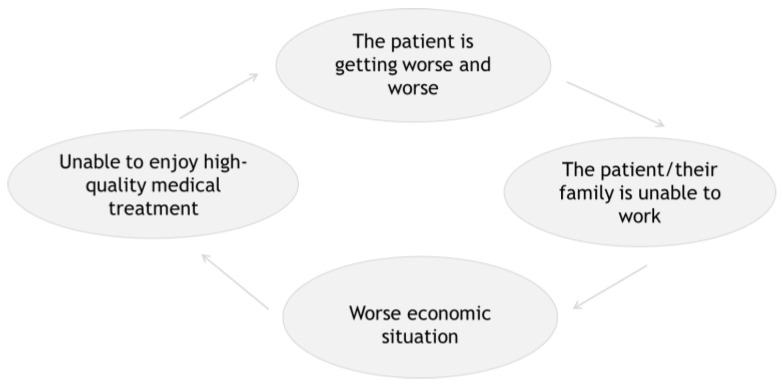
Flow chart of mutual relationship of SES and health level.

**Table 1 ijerph-19-01629-t001:** Descriptive statistics for continuous variables.

	Male				Female			
Variable	Mean	Std. Dev.	Min	Max	Mean	Std. Dev.	Min	Max
Out-of-pocket medical Expenses	124,793 (JPY)	165,468	0 (JPY)	1,500,000 (JPY)	158,653 (JPY)	137,758	0 (JPY)	800,000 (JPY)
Household income	5,804,537 (JPY)	3,892,684	0 (JPY)	25,000,000 (JPY)	4,723,366 (JPY)	6,063,347	0 (JPY)	65,000,000 (JPY)
HbA1c	7.34	1.29	4	10	7.26	1.09	4	10
	**Male**				**Female**			
	**Observations**		**Proportion (%)**		**Observations**		**Proportion (%)**	
Complication								
No complication	148		72.20		171		83.41	
One complication	42		20.49		23		11.22	
Two or more complication	15		7.32		11		5.37	
Time discount rate								
−5–0%	40		19.51		35		17.07	
2–6%	47		22.93		27		13.17	
10%	39		19.02		58		28.29	
20%	30		14.63		34		16.59	
40%	49		23.90		51		24.88	
The frequency of outpatient visit								
Once or few times per week	26		12.68		13		6.34	
Once per month	145		70.73		159		77.56	
Once or few times per year	34		16.59		33		16.10	
Onset timing								
0–9 years old	27		13.17		43		20.98	
10–19 years old	28		13.66		37		18.05	
20–29 years old	34		16.59		41		20.00	
30–39 years old	54		26.34		35		17.07	
40–49 years old	37		18.05		25		12.20	
Over 50 years old	25		12.20		24		11.71	
Care dummy by age range								
10–19 years old								
working caregiver	17		65.38		18		64.29	
“potential full-time caregiver”	9		34.62		10		35.71	
20–39 years old								
working caregiver	36		83.72		69		93.24	
“potential full-time caregiver”	7		16.28		5		6.76	
40–49 years old								
working caregiver	40		67.8		52		92.86	
“potential full-time caregiver”	19		32.2		4		7.14	
50 years and older								
working caregiver	47		61.04		35		74.47	
“potential full-time caregiver”	30		38.96		12		25.53	
Government aid dummy								
Not receiving	181		88.29		179		87.32	
Receiving	24		11.71		26		12.68	
Subtotal	205				205			

**Table 2 ijerph-19-01629-t002:** Estimated out-of-pocket medical expenses.

	Model 1	Model 2	Model 3
		Male	Female
Variables	Medical Expenditure	Medical Expenditure	Medical Expenditure
Household income	0.00561 ***	0.00273	0.00706 ***
	(0.00149)	(0.00334)	(0.00148)
Onset timing			
0–9 years old	3701	–41,421	1216
	(28,874)	(52,926)	(34,917)
10–19 years old	34,832	71,869 *	–12,601
	(26,171)	(43,319)	(32,339)
20–29 years old	23,950	–920.5	24,830
	(23,789)	(38,377)	(29,457)
40–49 years old	–2601	23,819	–67,279 *
	(25,878)	(37,475)	(37,390)
50 years and older	–27,649	21,493	–124,258 **
	(30,985)	(45,317)	(47,761)
Frequency of outpatient visits			
Once or a few times per week	–46,932*	–63,130 *	–41,855
	(25,387)	(36,250)	(36,989)
Once or a few times per year	–31,422	–52,121	–29,499
	(21,663)	(36,812)	(25,981)
Number of complications			
One complication	–7156	–1277	14,039
	(22,245)	(32,937)	(31,671)
Two or more complications	–2198	30,028	–54,915
	(31,526)	(48,585)	(43,088)
Time discount rate			
–5–0%	25,656	31,778	10,687
	(22,883)	(36,437)	(27,938)
2–6%	–17,564	–1836	–43,965
	(23,688)	(36,392)	(29,975)
10%	4667	–6693	36,125
	(21,372)	(36,253)	(28,111)
20%	52,941 **	73,016 *	–13,180
	(24,145)	(39,516)	(25,502)
Female dummy	28,553 *		
	(15,645)		
Care dummy by age range(Reference: 50 years and older * working caregiver)			
10–19 years old*working caregiver	–21,074	179,956	–247,902 **
	(81,679)	(127,128)	(102,972)
10–19 years old and “potential full-time caregiver”	–5428	229,617 *	–256,733 **
	(82,134)	(128,330)	(103,358)
20–39 years old and working caregiver	–7959	51,982	–78,389 **
	(27,496)	(44,061)	(38,313)
20–39 years old and “potential full-time caregiver”	–17,017	–22,025	–16,600
	(47,974)	(70,294)	(65,612)
40–49 years old and working caregiver	–46,630 *	–25,038	–90,744 ***
	(25,310)	(38,186)	(34,768)
40–49 years old and “potential full-time caregiver”	–52,497	–27,051	–101,597
	(36,070)	(46,253)	(72,257)
50 years and older and working caregiver	0	0	0
	(0)	(0)	(0)
50 years and older and “potential full-time caregiver”	–39,671	–4437	–47,351
	(28,323)	(40,125)	(45,574)
Government aid dummy	–132,319 *	–276,758 **	25,930
	(76,657)	(119,933)	(93,531)
Constant	129,261 ***	104,400 **	240,710 ***
	(28,614)	(42,619)	(42,346)
Observations	410	205	205
R-squared	0.154	0.159	0.283
Prob > F	0.0000	0.0588	0.0000
Standard errors in parentheses			

*** *p* < 0.01, ** *p* < 0.05, * *p* < 0.1.

**Table 3 ijerph-19-01629-t003:** Marginal effect on the out-of-pocket medical expenditure of care dummy by age.

		Male	Female
		Marginal Value	Marginal Value
10–19 years old	1	60,352.9	48,911.1 **
	2	55,316.7	44,951.2
	3	110,014.0 *	40,079.9
	4	104,977.8 *	36,120.0
20–39 years old	1	174,417.1 ***	189,681.2 ***
	2	195,349.7 ***	146,467.1 ***
	3	100,410.3	251,470.1 ***
	4	121,342.9 **	208,256.0 ***
40–49 years old	1	109,392.9 ***	165,373.6 ***
	2	106,330.2 ***	177,323.1 ***
	3	107,379.3 *	154,520.5 **
	4	104,316.6 *	166,470.0 **
50 years and older	1	134,981.9 ***	188,648.4 ***
	2	130,486.4 ***	154,434.0 ***
	3	130,544.6 ***	141,297.7 ***
	4	126,049.2 ***	107,083.3 **

Notes: *** *p* < 0.01, ** *p* < 0.05, * *p* < 0.1. Case 1: Predicted value when applying the effect of working if the caregiver is working. Case 2: Predicted value when applying the effect of being a “potential full-time caregiver” when caregiver is working. Case 3: Predicted value when applying the effect of working when the caregiver is a “potential full-time caregiver”. Case 4: Predicted value when applying the effect as a “potential full-time caregiver” when the caregiver is a “potential full-time caregiver”.

**Table 4 ijerph-19-01629-t004:** Regression results for HbA1c (males).

	Model 4	Model 5
Variables	HbA1c	HbA1c
Household income	3.72 × 10^–12^	9.12 × 10^–10^
	(2.62 × 10^–8^)	(2.50 × 10^–8^)
Onset timing		
0–9 years old	–0.210	–0.276
	(0.415)	(0.379)
10–19 years old	0.248	0.182
	(0.340)	(0.328)
20–29 years old	–0.405	–0.405
	(0.301)	(0.286)
30–49 years old	–0.0212	–0.00232
	(0.294)	(0.277)
50 years and older	0.250	0.152
	(0.355)	(0.314)
Frequency of outpatient visits		
Once or a few times per week/month	–0.229	–0.224
	(0.284)	(0.278)
Once or a few times per year	0.258	0.274
	(0.289)	(0.284)
Complications		
One complication	–0.662 **	–0.630 **
	(0.258)	(0.250)
Two or more complications	0.425	0.434
	(0.381)	(0.376)
Time discount rate		
–5–0%	–0.912 ***	–0.912 ***
	(0.286)	(0.277)
2–6%	–0.645 **	–0.615 **
	(0.285)	(0.278)
10%	–0.474 *	–0.458 *
	(0.284)	(0.276)
20%	–0.554 *	–0.552 *
	(0.310)	(0.303)
“potential full-time caregiver” (reference:50 years and older and working caregiver)		
“potential full-time caregiver” dummy		–0.324 *
		(0.195)
10–19 years old and working caregiver	–1.299	
	(0.997)	
10–19 years old and “potential full-time caregiver”	–1.206	
	(1.006)	
20–39 years old and working caregiver	0.0241	
	(0.346)	
20–39 years old and “potential full-time caregiver”	–0.241	
	(0.551)	
40–49 years old and working caregiver	0.124	
	(0.299)	
40–49 years old and “potential full-time caregiver”	–0.135	
	(0.363)	
50 years and older and working caregiver	0	
	(0)	
50 years and older and “potential full-time caregiver”	–0.480	
	(0.315)	
Government aid	1.754 *	0.632 *
	(0.940)	(0.373)
Constant	7.955 ***	7.988 ***
	(0.334)	(0.280)
Observations	205	205
R-squared	0.160	0.145
Prob > F	0.0565	0.0152
Standard errors in parentheses		

*** *p* < 0.01, ** *p* < 0.05, * *p* < 0.1.

**Table 5 ijerph-19-01629-t005:** Regression results for HbA1c (females).

	Model 6	Model 7
Variables	HbA1c	HbA1c
Household income	–6.31 × 10^–9^	–6.11 × 10^–9^
	(1.25 × 10^–8^)	(1.25 × 10^–8^)
Onset timing		
0–9 years old	0.362	0.401
	(0.293)	(0.287)
10–19 years old	0.511 *	0.504 *
	(0.271)	(0.266)
20–29 years old	0.0253	0.0194
	(0.247)	(0.248)
30–49 years old	0.540*	0.453
	(0.314)	(0.280)
50 years and older	0.338	–0.00839
	(0.401)	(0.300)
Frequency of outpatient visits		
Once or a few times per week/month	–0.803 **	–0.793 **
	(0.310)	(0.311)
Once or a few times per year	–0.502 **	–0.477 **
	(0.218)	(0.214)
Complications		
One complication	0.207	0.207
	(0.266)	(0.260)
Two or more complications	0.636 *	0.691 **
	(0.362)	(0.347)
Time discount rate		
–5–0%	0.250	0.246
	(0.234)	(0.235)
2–6%	0.204	0.134
	(0.252)	(0.249)
10%	0.225	0.179
	(0.236)	(0.233)
20%	0.0619	0.0985
	(0.214)	(0.209)
“potential full-time caregiver” (reference: 50 years and older * working caregiver)		
Full- time dummy		0.121
		(0.223)
10–19 years old and working caregiver	–1.309	
	(0.864)	
10–19 years old and “potential full-time caregiver”	–0.598	
	(0.867)	
20–39 years old and working caregiver	0.184	
	(0.321)	
20–39 years old and “potential full-time caregiver”	0.987*	
	(0.551)	
40–49 years old and working caregiver	0.216	
	(0.292)	
40–49 years old and “potential full-time caregiver”	–0.183	
	(0.606)	
50 years and older and working caregiver	0	
	(0)	
50 years and older and “potential full-time caregiver”	–0.348	
	(0.382)	
Government aid	1.403 *	0.116
	(0.785)	(0.283)
Constant	6.811 ***	6.974 ***
	(0.355)	(0.239)
Observations	205	205
R-squared	0.196	0.153
Prob > F	0.0064	0.0092
Standard errors in parentheses		

*** *p* < 0.01, ** *p* < 0.05, * *p* < 0.1.

**Table 6 ijerph-19-01629-t006:** Marginal effect on HbA1c of care dummy by age range.

		Male	Female
	Case	Marginal Value	Marginal Value
10–19 years old	1	7.97 ***	7.33 ***
	2	7.73 ***	7.19 ***
	3	7.64 ***	8.04 ***
	4	7.41 ***	7.90 ***
20–39 years old	1	7.27 ***	7.22 ***
	2	7.41 ***	7.40 ***
	3	6.94 ***	8.02 ***
	4	7.09 ***	8.20 ***
40–49 years old	1	7.38 ***	7.35 ***
	2	7.39 ***	7.15 ***
	3	7.06 ***	6.95 ***
	4	7.06 ***	6.75 ***
50 years and older	1	7.33 ***	7.09 ***
	2	7.54 ***	7.10 ***
	3	7.01 ***	6.74 ***
	4	7.22 ***	6.75 ***

Notes: *** *p* < 0.01. Case 1: Predicted value when applying the effect of working if the caregiver is working. Case 2: Predicted value when applying the effect of being a “potential full-time caregiver” when caregiver is working. Case 3: Predicted value when applying the effect of working when the caregiver is a “potential full-time caregiver”. Case 4: Predicted value when applying the effect as a “potential full-time caregiver” when the caregiver is a “potential full-time caregiver”.

**Table 7 ijerph-19-01629-t007:** Marginal effect on HbA1c of care dummy by age range and government aid.

		Male	Female
		Marginal Value	Marginal Value
10–19 years old	1	6.11 ***	6.01 ***
	2	6.00 ***	6.64 ***
	3	7.87 ***	7.41 ***
	4	7.75 ***	8.04 ***
20–39 years old	1	7.28 ***	7.22 ***
	2	7.14 ***	8.20 ***
	3	9.03 ***	8.62 ***
	4	8.90 ***	9.60 ***
40–49 years old	1	7.48 ***	7.35 ***
	2	7.21 ***	6.75 ***
	3	9.23 ***	8.75 ***
	4	8.96 ***	8.15 ***
50 years and older	1	7.32 ***	7.09 ***
	2	7.07 ***	6.75 ***
	3	9.07 ***	8.49 ***
	4	8.82 ***	8.15 ***
Total age	1	7.21 ***	7.23 ***
	2	6.97 ***	7.32 ***
	3	8.96 ***	7.35 ***
	4	8.72 ***	7.44 ***

Notes: *** *p* < 0.01. Case 1: Predicted value when applying the effect of a caregiver working in the absence of government aid. Case 2: Predicted value when the effect of the caregiver being a “potential full-time caregiver” is applied in the absence of government aid. Case 3: Predicted value when the effect of the caregiver working is applied in the presence of government aid. Case 4: Predicted value when caregiver is a “potential full-time caregiver” in the absence of government aid.

## Data Availability

The data that support the findings of this study are available on request from the corresponding author. The data are not publicly available due to privacy reasons.

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
