# Peer review of "Full-Time or Working Caregiver? A Health Economics Perspective on the Supply of Care for Type 1 Diabetes Patients"

_ijerph, 2022, doi:10.3390/ijerph19031629_

Round 1
Reviewer 1 Report
Dear Author,
he topic seems to be very interesting and practical one. I am involved in this topic as a scientist and also my family could be as case in that issue. So I am really convinces to publishing a text but after necessary correction. Most of weak point are going to the manner of presentation:
1. Most of the article looks like a supplement due to the large number of tables too narrowly described and placed one after the other. This part requires a broader description of threads, ex. which information will be announced in the next table.
2. Too pressed information in lines 146 -166 - or maybe to Discussion part?
Results....the same problems like with Materials....to much tabel (numbers) on the pages...without appropriate comments.
3. You need to better title the table, columns, lack of currency entry...
4. Discussion part has no discussion with the results of other researchers;
5. The conclusions reiterated what we had already read twice in the text;
6. The style of citation in a pure text must be corrected...see line 50, 59, etc.
Author Response
Dear Referees,
Let me express to thank you for your detailed comments warmly. I reply to each referee’s comments as follows. I am incredibly grateful to you, and I hope that you will find answers to be satisfactory. In any case, thank you for your comments, as my paper has been improved considerably. 
The author.
Replies of the authors to referee # 1
he topic seems to be very interesting and practical one. I am involved in this topic as a scientist and also my family could be as case in that issue. So I am really convinces to publishing a text but after necessary correction. Most of weak point are going to the manner of presentation:
- Most of the article looks like a supplement due to the large number of tables too narrowly described and placed one after the other. This part requires a broader description of threads, ex. which information will be announced in the next table.
Reply
Thank you so much for your comments. I added explanations especially carefully for descriptive statistics, as is indicated by the referee. Please find the modified sentences as following in the line 154 to 208.
We briefly describe the epidemiological findings arising from the descriptive statis-tics, which is shown in table 1. Table 1 contains 2 brackets; one is for continuous variables and the other is categorical variables.
The upper bracket of table 1 contains the descriptive statistics for continuous variables. The explanation of each continuous variable is below. Firstly, out-of-pocket medical expenses are the amount of out-of-pocket expenses for IDDM treatment and complications during 2015. The amount paid at the medical institution, even if patients received a refund due to "High-cost Medical Expense Benefit". This "High-cost Medical Expense Benefit" can be claimed if the cost-bearing limit of medical expenses exceeds the specified amount for the month. The total amount (30% of medical costs) is paid first, later receiving a refund. In the questionnaire, we asked, "Please provide the amount of your out-of-pocket expenses for IDDM treatment and complications over the last year. (*Please enter the amount you paid at the medical institution, even if you received a refund due to high-cost medical care expenses, etc.)". Secondly, household income is the amount of household income in 2015. Lastly, HbA1c is the latest HbA1c value (NGSP value) and we asked, "Please provide us with the most recent HbA1c value (NGSP value) of patient".
The lower bracket of table 1, on the other hand, contains the descriptive statistics for categorical variables.
In questionnaire, we asked whether respondents have following complications; diabetic retinopathy, diabetic nephropathy, neuropathy, gangrene, coronary artery disease, cerebrovascular disease, other. As for complications, most patients do not experience any complications.
Time discount rare which is obtained from the question "How much would you be happy to receive as a minimum after 13 months instead of 10,000 yen after one month?" shows the discount placed on returns receivable or costs payable in the future. The use of a high dis-count rate implies that people put less weight on the future and they are impatience. Most of time discount rate is under 10%, which means they put high weight on the future.
As for onset timing varies, with around 48.7% of patients experiencing the onset of T1DM after age 30, which is consistent with [33]. Those who have onset over 20 are unable to capture by previous survey in Japan, e.g. [34] and [35], which aims to track juvenile on-set and “Grant-in-Aid Program for Chronic Diseases in Childhood” users.
The most frequent use of outpatient services is once a month.
Then we shift to the explanation of the variable to capture caregiver of patients. As for juvenile patient, it has been pointed out that 97.5% of children with disabilities at home are primarily cared for by their mothers, and that children with medical care needs place a heavy load on mothers [36],[37]. It is clarified patient with spouse demands newly developed medical equipment in [27], then we focus patient’s mother and spouse as caregivers, which is categorized “care dummy” in this article.
We specify the care dummy by age range. If the patient's mother or spouse, for both wife and husband, is a full-time homemaker or not working, the caregiver dummy takes a value of one, otherwise zero. Hereinafter we define this caregiver as a “potential full-time caregiver”.
In the field of labour economics, there is a socio-economic bias when engagement of a patient’s mother or wife in paid work is related to their own and husband's economic level, a tendency recognized as "Douglas–Arisawa's second law". The rule of thumb is that the higher the head of household's income, the less likely the spouse has a job. However, theoretically, we can dismiss such a bias because of the random onset of T1DM. As is can be seen from the descriptive statistics, the labour status of the mother or spouse of a patient under 20 years old indicates low labour force participation. For single patients under 20, 63.46% of mothers work. This result is consistent with previous studies such as [30].
Lastly, we also focus those who receive governmental aid. This “government aid dummy” takes a value of one if the patient under 20 years old receives at least one of the "Grant-in-Aid Program for Chronic Diseases in Childhood" or "Special Child Rearing Allowance", otherwise zero. For example, for adult recipients who have received some government aid in the past, the dummy variable has a value of zero.
- Too pressed information in lines 146 -166 - or maybe to Discussion part?
Results....the same problems like with Materials....to much tabel (numbers) on the pages...without appropriate comments.
Reply
Thank you so much for comments. As for “Too pressed information in lines 146 -166 - or maybe to Discussion part?”, We added more precise explanations in line 154 to 195 as following.
We briefly describe the epidemiological findings arising from the descriptive statis-tics, which is shown in table 1. Table 1 contains 2 brackets; one is for continuous variables and the other is categorical variables.
The upper bracket of table 1 contains the descriptive statistics for continuous variables. The explanation of each continuous variable is below. Firstly, out-of-pocket medical expenses are the amount of out-of-pocket expenses for IDDM treatment and complications during 2015. The amount paid at the medical institution, even if patients received a refund due to "High-cost Medical Expense Benefit". This "High-cost Medical Expense Benefit" can be claimed if the cost-bearing limit of medical expenses exceeds the specified amount for the month. The total amount (30% of medical costs) is paid first, later receiving a refund. In the questionnaire, we asked, "Please provide the amount of your out-of-pocket expenses for IDDM treatment and complications over the last year. (*Please enter the amount you paid at the medical institution, even if you received a refund due to high-cost medical care expenses, etc.)". Secondly, household income is the amount of household income in 2015. Lastly, HbA1c is the latest HbA1c value (NGSP value) and we asked, "Please provide us with the most recent HbA1c value (NGSP value) of patient".
The lower bracket of table 1, on the other hand, contains the descriptive statistics for categorical variables.
In questionnaire, we asked whether respondents have following complications; diabetic retinopathy, diabetic nephropathy, neuropathy, gangrene, coronary artery disease, cerebrovascular disease, other. As for complications, most patients do not experience any complications.
Time discount rare which is obtained from the question "How much would you be happy to receive as a minimum after 13 months instead of 10,000 yen after one month?" shows the discount placed on returns receivable or costs payable in the future. The use of a high dis-count rate implies that people put less weight on the future and they are impatience. Most of time discount rate is under 10%, which means they put high weight on the future.
As for onset timing varies, with around 48.7% of patients experiencing the onset of T1DM after age 30, which is consistent with [33]. Those who have onset over 20 are unable to capture by previous survey in Japan, e.g. [34] and [35], which aims to track juvenile on-set and “Grant-in-Aid Program for Chronic Diseases in Childhood” users.
The most frequent use of outpatient services is once a month.
Then we shift to the explanation of the variable to capture caregiver of patients. As for juvenile patient, it has been pointed out that 97.5% of children with disabilities at home are primarily cared for by their mothers, and that children with medical care needs place a heavy load on mothers [36],[37]. It is clarified patient with spouse demands newly developed medical equipment in [27], then we focus patient’s mother and spouse as caregivers, which is categorized “care dummy” in this article.
We specify the care dummy by age range. If the patient's mother or spouse, for both wife and husband, is a full-time homemaker or not working, the caregiver dummy takes a value of one, otherwise zero. Hereinafter we define this caregiver as a “potential full-time caregiver”.
As for “Results....the same problems like with Materials....to much tabel (numbers) on the pages...without appropriate comments.”, we have increased the amount of text and explanation throughout.
- You need to better title the table, columns, lack of currency entry...
Reply
Thank you so much for comments. We modified the title and added the explanation of currency.
- Discussion part has no discussion with the results of other researchers;
Reply
Thank you so much for your comments. Thanks to your advice, I found important literature and modified as following in the line 376 to 390.
However, this result does not always valid in the case of long-term data analysis, since it is referred that families that experienced reduced financial resources because of lost in-come from either a cut in work hours or having to stop work perceived more significant financial burden, and often responded that they needed additional income to cover medical expenditures [38].
The expenditure amount may be relatively small compared to other household expenses; families with children who need special health care may be financially stretched because of many years of persistently costly care resulting in lower expenditures causing the perceptions of problems [39].
It is also referred that meeting the fiscal demands of health care expenditures with decreased financial resources may result in families selling assets, taking out loans or mortgages, using savings, or seeking additional employment [40]. Re-establishing financial well-being may be hindered by decreased employability and loss of career advancement and mobility caused by being out of the workplace for any extended period to care for the child [41].
- The conclusions reiterated what we had already read twice in the text;
Reply
Thank you so much for your comment. We modified the conclusion sections as following.
This study focused on the impact of SES on health status and vice versa, with particular attention to differences in the SES of caregivers of patients with T1DM in Japan. Using an original survey conducted in 2016 targeted at Japanese T1DM patients, we studied the following three points:
- We run the regression on out-of-pocket medical expenditure and HbA1c based on the patient’s and caregiver's socio-economic factors, especially the caregiver’s labour status.
- Based on the above results, we estimated counterfactual out-of-pocket medical expenditure and HbA1c.
- We also estimated a hypothetical value of the impact of either health care funding or caregiver presence on blood glucose levels.
When we looked at differences in caregiver employment status, we found that the effects varied by sex and patient age. The overall finding is that counterfactual HbA1c is more influenced by caregiver employment status than by government funding. For women under 40 years old, HbA1c was lower when the caregiver was employed. For the rest of the population, i.e., men and women 40 and older having an unemployed caregiver resulted in a lower HbA1c. As discussed, this conclusion is not for the purpose of denying the benefit of government aid such as the "Grant-in-Aid Program for Chronic Diseases in Childhood" and "Special Child Rearing Allowance". These governmental aids are available only to paediatric patients and have time restrictions. In future research, we will assess how continuous economic policy relating to T1DM patients would be appropriate.
The conclusions of this paper do not negate the need for health care funding. This is because medical devices such as SAPs and CSIIs, which lower HbA1c, have either not yet penetrated the Japanese health care system or are not used because they interfere with daily life and require blood glucose control by oneself or family members. Therefore, if patients, their families, and doctors are informed about these medical devices, and if the quality of medical devices further improves, it is expected to be more effective than the presence of caregivers. Further, while there is a need for additional analysis, it seems that there is also a need for economic policies such as employment and wages for people with T1DM and their families, rather than time-limited health care subsidies.
- The style of citation in a pure text must be corrected...see line 50, 59, etc.
Reply
Thank you so much for detailed comments. I modified the citation as followings;
Before (line 50)
[26] clarified T1DM in childhood has a negative effect on labour market outcomes later in life, which means the causality instead runs from health to socio-economic status.
After
Previous research clarified that T1DM in childhood has a negative effect on labour market outcomes later in life, which means the causality instead runs from health to SES [26].
Before (line 59)
[27] found that parental and spousal recommendations are important for the demand for better medical equipment not only for paediatric patients but also for adult patients.
After
Literature found that parental and spousal recommendations are important in the demand for better medical equipment, not only for paediatric patients but also adult patients [27].

Reviewer 2 Report
The manuscript is very hard to read. The sentence don't always logically flow. There is no clear research question and it seems like the authors may have tried multiple different things at once, without distinguishing how variables affecting the outcomes of interest (i.e. health expenditure and Hba1c level). It does not seem appropriate to consider wife of someone living with T1D and mother of a child with T1D. I also do not understand why authors assume that an adult needs a caregiver for the management of T1D (unless there are other issues at stake). It is very confusing to analyze everything together. Also, why do authors assume that mothers/wives are the caregiver? What about fathers/husbands?
Author Response
Dear Referees,
Let me express to thank you for your detailed comments warmly. I reply to each referee’s comments as follows. I am incredibly grateful to you, and I hope that you will find answers to be satisfactory. In any case, thank you for your comments, as my paper has been improved considerably. 
The author.
Replies of the authors to referee # 2
The manuscript is very hard to read. The sentence don't always logically flow.
Reply:
Thank you very much for your comment. We have revised my English completely.
There is no clear research question and it seems like the authors may have tried multiple different things at once, without distinguishing how variables affecting the outcomes of interest (i.e. health expenditure and Hba1c level).
Reply:
Thank you very much for your comment. In pp. 2-3, we rewrote the research question to emphasize as following.
Figure 1 illustrates the flowchart of causality running from SES to health level and health level to SES. As shown in figure 1, does the inactivity of the patient and their family's work lead to a lack of access to high-quality medical treatment? or does it lead to maintain health condition by concentrating on care? To solve this puzzle, we need to focus on the labour status of the patient and their family.
Figure 1. Flow chart of mutual relationship of SES and health level.
In this paper, we attempt to evaluate the following hypothesis. By mothers becoming full-time homemakers or caretakers, are medical expenditures reduced and is the health level represented by HbA1c improved? To respond to this question, we employ counterfactual medical expenditure and HbA1c, shedding light on labour status, especially that of the patient's caregiver, using regression analysis.
It does not seem appropriate to consider wife of someone living with T1D and mother of a child with T1D. I also do not understand why authors assume that an adult needs a caregiver for the management of T1D (unless there are other issues at stake). It is very confusing to analyze everything together. Also, why do authors assume that mothers/wives are the caregiver? What about fathers/husbands?
Reply:
Thank you so much for your comments. We don't intend to specify female caregivers; however, we insist to focus on caregivers of mother and spouse (male and female). We added following sentence in the line 186 to 191.
As for juvenile patient, it has been pointed out that 97.5% of children with disabilities at home are primarily cared for by their mothers, and that children with medical care needs place a heavy load on mothers [36],[37]. It is clarified patient with spouse demands newly developed medical equipment in [27], then we focus patient’s mother and spouse as caregivers, which is categorized “care dummy” in this article.
And we modified the following explanation in the line 64 to line 71.
Focusing on Japan, the number of children requiring daily medical care (usually called "medical care children" in Japan) is increasing each year, with 18,272 cases in 2016 [28]. Especially for parents are the primary caregivers of children in medical care, and the load on mothers is particularly high. Despite this, in Japan, most studies have focused on the employment of patients themselves and there are fewer studies concerning the employment of the caregivers of paediatric patients than in other countries. There is a lack of basic national data on the employment status of families with children in medical care. There are no concrete measures to support parents' employment.

Reviewer 3 Report
The article deals with an interesting and worthy subject to be published but not in the present form.
In fact, in my opinion there are improvements that could make the work more understandable for readers.
The materials and methods section needs to be reorganized. I suggest dividing it into sub-chapters:
one in which the design, the setting and the population under study are explained;
one in which the outcomes of interest are described (I, for example, did not understand how the HbA1c data was collected);
one dedicated to statistical analysis: in fact it is necessary to better describe the methodology without taking some important steps for granted. The methodology used for descriptive statistics must also be explained on the basis of the distribution of the variables (mean or median use, use of standard deviations or interquartile ranges?). As regards the inferential statistics and the chosen regression, it is necessary to declare whether the assumptions are respected. for example, to perform a linear regression it is necessary that the variables respect some assumptions (please clarify this methodological aspect!).
In the version I have seen, table 1 is not clearly legible. It is probably necessary to divide it into 2, that is, one above the line 145 and one after that line. It is necessary to know the numbers entered what they refer to by indicating N for a number,% for percentages and so on.
This is important for all other tables, even for regression ones (what is a coefficient indicated? And between brackets?) Please arrange the tables so that it is clearly legible making the values traceable to measures without taking for granted.
In my opinion a work should be cited indicating the name of the first author and indicating immediately after the bibliographic reference. For example in line 59 I would write Sakoda et al. [27] etc.
In addition to these ideas that enter into the merits of the presentation, there are also some typos scattered throughout the text:
line 47 and line 74: square brackets are enough.
lines 110 and 158: capital letter after the period
in line 273 I don't understand the meaning of (1).
Author Response
Dear Referees,
Let me express to thank you for your detailed comments warmly. I reply to each referee’s comments as follows. I am incredibly grateful to you, and I hope that you will find answers to be satisfactory. In any case, thank you for your comments, as my paper has been improved considerably. 
The author.
Replies of the authors to referee # 3
The article deals with an interesting and worthy subject to be published but not in the present form.
In fact, in my opinion there are improvements that could make the work more understandable for readers.
The materials and methods section needs to be reorganized. I suggest dividing it into sub-chapters:
one in which the design, the setting and the population under study are explained;
one in which the outcomes of interest are described (I, for example, did not understand how the HbA1c data was collected);
one dedicated to statistical analysis: in fact it is necessary to better describe the methodology without taking some important steps for granted. The methodology used for descriptive statistics must also be explained on the basis of the distribution of the variables (mean or median use, use of standard deviations or interquartile ranges?). As regards the inferential statistics and the chosen regression, it is necessary to declare whether the assumptions are respected. for example, to perform a linear regression it is necessary that the variables respect some assumptions (please clarify this methodological aspect!).
Reply:
Thank you so much for your comments. Based on your sincere advice, I divided section 2 into 2 for better describe. One is material section to explain the materials, e.g. which the design, the setting and the population under the study. The other is methods section to explain design and statistical analysis. I hope my paper has been improved considerably.
As for collecting HbA1c, I also added the explanation as well as following in the line 169 and 170.
HbA1c is the latest HbA1c value (NGSP value) and we asked, "Please provide us with the most recent HbA1c value (NGSP value) of patient".
In the version I have seen, table 1 is not clearly legible. It is probably necessary to divide it into 2, that is, one above the line 145 and one after that line. It is necessary to know the numbers entered what they refer to by indicating N for a number,% for percentages and so on.
Reply:
Thank you so much for your comments. Along with your suggestion, I added the explanation in the table 1 and under the line 156 as following.
The upper bracket of table 1 contains the descriptive statistics for continuous variables. The explanation of each continuous variable is below. Firstly, out-of-pocket medical expenses are the amount of out-of-pocket expenses for IDDM treatment and complications during 2015. The amount paid at the medical institution, even if patients received a refund due to "High-cost Medical Expense Benefit". This "High-cost Medical Expense Benefit" can be claimed if the cost-bearing limit of medical expenses exceeds the specified amount for the month. The total amount (30% of medical costs) is paid first, later receiving a refund. In the questionnaire, we asked, "Please provide the amount of your out-of-pocket expenses for IDDM treatment and complications over the last year. (*Please enter the amount you paid at the medical institution, even if you received a refund due to high-cost medical care expenses, etc.)". Secondly, household income is the amount of household income in 2015. Lastly, HbA1c is the latest HbA1c value (NGSP value) and we asked, "Please provide us with the most recent HbA1c value (NGSP value) of patient".
The lower bracket of table 1, on the other hand, contains the descriptive statistics for categorical variables.
In questionnaire, we asked whether respondents have following complications; diabetic retinopathy, diabetic nephropathy, neuropathy, gangrene, coronary artery disease, cerebrovascular disease, other. As for complications, most patients do not experience any complications.
Time discount rare which is obtained from the question "How much would you be happy to receive as a minimum after 13 months instead of 10,000 yen after one month?" shows the discount placed on returns receivable or costs payable in the future. The use of a high dis-count rate implies that people put less weight on the future and they are impatience. Most of time discount rate is under 10%, which means they put high weight on the future.
As for onset timing varies, with around 48.7% of patients experiencing the onset of T1DM after age 30, which is consistent with [33]. Those who have onset over 20 are unable to capture by previous survey in Japan, e.g. [34] and [35], which aims to track juvenile on-set and “Grant-in-Aid Program for Chronic Diseases in Childhood” users.
The most frequent use of outpatient services is once a month.
Then we shift to the explanation of the variable to capture caregiver of patients. As for juvenile patient, it has been pointed out that 97.5% of children with disabilities at home are primarily cared for by their mothers, and that children with medical care needs place a heavy load on mothers [36],[37]. It is clarified patient with spouse demands newly developed medical equipment in [27], then we focus patient’s mother and spouse as caregivers, which is categorized “care dummy” in this article.
We specify the care dummy by age range. If the patient's mother or spouse, for both wife and husband, is a full-time homemaker or not working, the caregiver dummy takes a value of one, otherwise zero. Hereinafter we define this caregiver as a “potential full-time caregiver”.
In the field of labour economics, there is a socio-economic bias when engagement of a patient’s mother or wife in paid work is related to their own and husband's economic level, a tendency recognized as "Douglas–Arisawa's second law". The rule of thumb is that the higher the head of household's income, the less likely the spouse has a job. However, theoretically, we can dismiss such a bias because of the random onset of T1DM. As is can be seen from the descriptive statistics, the labour status of the mother or spouse of a patient under 20 years old indicates low labour force participation. For single patients under 20, 63.46% of mothers work. This result is consistent with previous studies such as [30].
Lastly, we also focus those who receive governmental aid. This “government aid dummy” takes a value of one if the patient under 20 years old receives at least one of the "Grant-in-Aid Program for Chronic Diseases in Childhood" or "Special Child Rearing Allowance", otherwise zero. For example, for adult recipients who have received some government aid in the past, the dummy variable has a value of zero.
This is important for all other tables, even for regression ones (what is a coefficient indicated? And between brackets?) Please arrange the tables so that it is clearly legible making the values traceable to measures without taking for granted.
Reply:
Thank you so much for your comments. What is described in parentheses, as you mentioned as “between brackets” shows standard errors. it is described in the bottom of all regression results tables as “Standard errors in parentheses”. I’d most appreciate it if the revised manuscript is clearly legible for you.
In my opinion a work should be cited indicating the name of the first author and indicating immediately after the bibliographic reference. For example in line 59 I would write Sakoda et al. [27] etc.
Reply:
Thank you so much for your comments and I myself agree with you. This citation is along with the journal and publisher’s rule. I’d most appreciate it if you kindly accept the current citation style.
In addition to these ideas that enter into the merits of the presentation, there are also some typos scattered throughout the text:
Reply:
Thank you so much for your comments. I have revised my English completely. And let me reply to you for each comments.
line 47 and line 74: square brackets are enough.
lines 110 and 158: capital letter after the period
Reply:
Thank you so much for your comments. I modified line 158, however, line 110 refers to the company’s name, I’d most appreciate it if you accept it.
in line 273 I don't understand the meaning of (1).
Reply:
Thank you so much for your comments. I revised as following along with your advice.
We found that the impact of caregivers on counterfactual out-of-pocket medical expenditure was negative for younger age groups when the patient was female, i.e., out-of-pocket medical expenditure was lower. Based on the analysed counterfactual out-of-pocket medical expenditure, middle-aged patients with a full-time house-caregiver incur higher out-of-pocket medical expenditures. We explain this finding both directly and indirectly. Directly, a full-time house-caregiver may have more allowance to care about a patient's health condition and the frequency of outpatients. Indirectly, middle-aged patients are eager to maintain their health condition to support the patient's family. In terms of HbA1c, the full-time caregiver dummy is significant for all age groups for men and 20–39-year-old women. Based on these results, we estimated HbA1c and found that for women under 40 years old, having a “potential full-time caregiver” rather than government support was associated with lower HbA1c levels. Taken together with the regression analysis results, this effect is likely to be more vital for those aged between 20 and 40. In combination with the significant results for the regression analysis (Model 5) for all age groups of men, the presence of an unemployed person at home contributes to lowering HbA1c and maintaining health.
